# MiDAS: Multi-integrated Domain Adaptive Supervision for Fake News Detection

## Abstract

Covid-19 related misinformation and fake news, coined an 'infodemic', has dramatically increased over the past few years. This misinformation exhibits concept drift, where the distribution of fake news changes over time, reducing effectiveness of previously trained models for fake news detection. Given a set of fake news models trained on multiple domains, we propose an adaptive decision module to select the best-fit model for a new sample. We propose MiDAS, a multi-domain adaptative approach for fake news detection that ranks relevancy of existing models to new samples. MiDAS contains 2 components: a doman-invariant encoder, and an adaptive model selector. MiDAS integrates multiple pre-trained and fine-tuned models with their training data to create a domain-invariant representation. Then, MiDAS uses local Lipschitz smoothness of the invariant embedding space to estimate each model's relevance to a new sample. Higher ranked models provide predictions, and lower ranked models abstain. We evaluate MiDAS on generalization to drifted data with 9 fake news datasets, each obtained from different domains and modalities. MiDAS achieves new state-of-the-art performance on multi-domain adaptation for out-of-distribution fake news classification.

## 1 Introduction

The misinformation and fake news associated with the COVID-19 pandemic, called an 'infodemic' by WHO (Enders et al., 2020), have grown dramatically, and evolved with the pandemic. Fake news has eroded institutional trust (Ognyanova et al., 2020) and have increasingly negative impacts outside social communities (Quinn et al., 2021). The challenge is to filter active fake news campaigns while they are raging, just like today's online email spam filters, instead of offline, retrospective detection long after the campaigns have ended. We divide this challenge to detect fake news online into two parts: (1) the variety of data (both real and fake), and (2) the timeliness of data collection and processing (both real and fake). In this paper, we focus on the first (variety) part of challenge, with the timeliness (which depends on solutions to handle variety) in future work (Pu et al., 2020).

The infodemic, and fake news more generally, evolves with a growing variety of ephemeral topics and content, a phenomenon called real concept drift (Gama et al., 2014). However, the excellent results on single-domain classification (Chen et al., 2021), have generalization difficulties when applied to cross-domain experiments (Wahle et al., 2022; Suprem & Pu, 2022). A benchmark study over 15 language models shows reduced cross-domain fake news detection accuracy (Wahle et al., 2022). A generalization study in (Suprem & Pu, 2022) finds significant performance deterioration when models are used on unseen, non-overlapping datasets. Intuitively, it is entirely reasonable that state-of-the-art models trained on one dataset or time period will have reduced accuracy on future time periods. Real concept drift is introduced into fake news as content changes (Gama et al., 2014), camouflage (Shrestha & Spezzano, 2021), linguistic drift (Eisenstein et al., 2014), and adversarial adaptation by fake news producers when faced with debunking efforts such as CDC on the pandemic (Weinzierl et al., 2021).

To catch up with concept drift, the classification models need to be expanded to cover a wide variety of data sets (Li et al., 2021; Suprem & Pu, 2022; Kaliyar et al., 2021), or augmented with new knowledge on true novelty such as the appearance of the Omicron variant (Pu et al., 2020). In this paper, we assume the availability of domain-specific authorative sources such as CDC and WHO that provide trusted up-to-date information on the pandemic.

A key challenge of such multi-domain classifiers is a decision module to select the best-fit model amongst a set of existing models to classify new samples. This degree of knowledge is defined by the overlap between an unlabeled sample and existing models' training datasets (Suprem & Pu, 2022). Intuitively, a best-fit model better captures a sample point's neighborhood in its own training data Urner & Ben-David (2013); Chen et al. (2022).

**MiDAS.** We propose MɪDAS, a multi-domain adaptative approach for early fake news detection, with potential for online filtering. MɪDAS integrates multiple pre-trained and fine-tuned models along with their training data to create a domain-invariant representation. On this representation, MɪDAS uses a notion of local Lipschitz smoothness to estimate the overlap, and therefore relevancy, between a new sample and model training datasets. This overlap estimate is used to rank models on relevancy to the new sample. Then, MɪDAS selects the highest ranked model to perform classification. We evaluate MɪDAS on 9 fake news datasets obtained from different domains and modalities. We show new state-of-the-art performance on multi-domain adaptation for early fake news classification.

**Contributions.** Our contributions are as follows:

1. MɪDAS: a framework for adaptive model selection by using sample-to-data overlap to measure model relevancy
2. Experimental results of MɪDAS on 9 fake news datasets with state-of-the-art results using unsupervised domain adaptation.

## 2 RELATED WORK

### 2.1 MULTI-DOMAIN ADAPTATION

Domain adaptation maps a target domain into a source domain. This allows a classifier learned from the source domain to predict the target domain samples (Farahani et al., 2021). Some approaches focus on a domain invariant representation between source and target (Huang et al., 2021). Then, a new classifier can be trained on this invariant representation for both source and target samples. Domain invariance is scalable to multiple source domains by fusing their latent representations with an adversarial encoder-discriminator framework (Li et al., 2021). For multi-source domain adaptation (MDA), classifiers for each source have different weights: static weights using distance (Li et al., 2021) or per-sample weights on l2 norm (Suprem et al., 2020).

### 2.2 LABEL CONFIDENCE

Alongside domain adaptation, weak supervision (WS) is also common for propagating labels from source domains to a target domain (Ratner et al., 2017). Both approaches estimate labels closest to the true label of the target domain sample. This works with the assumption that the source domains or labeling functions, respectively, are correlated to the true labels due to expertise and domain knowledge. In each case, whether MDA or WS, domains or labeling functions need to be weighted to ensure reliance on the best-fit source. Snorkel, from (Ratner et al., 2017), uses expert labeling functions and weighs them on conditional independence. Similarly, approaches in (Chen et al., 2020; Fu et al., 2020) use coverage of expert foundation models and weigh on distance to embedded sample. EEWS from (Rühling Cachay et al., 2021) directly combines source data and labeling function in estimator parametrization to generate dynamic weights for each sample. MDA approaches weigh sources with weak supervision (Li et al., 2021), distance (Suprem & Pu, 2022), or as team-of-experts (Pu et al., 2020).

## 3 PROBLEM SETUP AND STRATEGY

Let there be $k$ source data domains, with labels $\{X_i, Y_i\}_{i=1}^k \in \{D\}_{i=1}^k$. Each of these source has an associated source model SM, with a total of $k$ SMs: $\{f_i\}_{i=1}^k$, where we have access to the training data $X_i$ and weights $w_i$. Each SM yields hidden embeddings through a feature extractor backbone, or foundation model (Bommasani et al., 2021). Embeddings are projected to class probabilities with any type of classification layer/module.

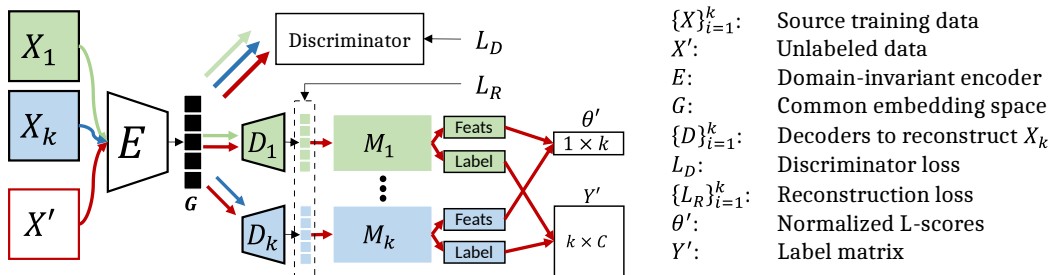

**Figure 1:** MIDAS architecture: The encoder generates a domain invariant representation. We add perturbation to this representation. Then, fine-tuned models $\{M\}_{(i=1)}^k$ process the sample and perturbations. After computing the local Lipschitz constant for each model, we can rank their L scores and select the best-fit model's label from the label matrix.

MIDAS adaptively weights the $k$ SM predictions for some unlabeled, potentially drifted target data domain $X'$. We accomplish this by using local embedding smoothness of the SMs as a proxy for model relevance to a sample $x' \in X'$. SMs are typically smooth in the embedding space (Urner & Ben-David, 2013); further, smoothness is correlated with local accuracy (Chen et al., 2022).

With MIDAS, we rank each $f_k$ on a smoothness measure around the embedding for $x'$, *i.e.* $f_k(x')$. Then, MIDAS can directly use the top-ranked $f_k$, or the smoothest $f_k$ under a smoothness threshold, as the best-fit relevant models for $x'$, with the remaining models abstaining. Because we are directly measuring smoothness on the embedding space, MIDAS can use already fine-tuned, state-of-the-art classifiers for each task, allowing off-the-shelf, plug-and-play usage. These classifiers are foundation models (Bommasani et al., 2021) that have been fine-tuned with architectural changes, learned weights, and hyperparameter tuning for their specific dataset.

There are two key challenges in MIDAS:

1. How do we compare smoothness of SMs that have been trained on different domains?
2. How can we measure the smoothness itself for unlabeled samples in the embedding space of SMs?

We address (1) with an encoder $E$ that generates a domain invariant representation on $\{X_i\}_{i=1}^k$. This unifies the data domains, allowing comparisons for different SMs to start from the same source domain. For (2), we extend the idea of local Lipschitz smoothness from (Chen et al., 2022) and (Urner & Ben-David, 2013) to randomized Lipschitz smoothness. In randomized Lipschitz smoothness, we randomly perturb $E(x')$, the domain invariant representation of $x'$. Then, we compute the local lipschitz constant $L$ on these perturbations $E(x') + \epsilon$, with respect to $E(x')$ to measure smoothness. This allows us to calculate an $L_k$ for each $f_k$ and use the local Lipschitz constant, a measure for the local smoothness, as a measure of relevancy.

## 4 MIDAS

We now describe MIDAS components and implementation details. The MIDAS architecture is shown in Figure 1. First, we cover the encoder-decoder framework to generate the domain invariant representations of source and target datasets. Then we present the randomized Lipschitz smoothness measure to generate SM relevancy rankings.

### 4.1 DOMAIN INVARIANT ENCODER

To compare different $f_k$ relevancy, we require a common source domain (Li et al., 2021). We achieve this with a single-encoder multiple-decoders design, where we have a single encoder to generate a domain invariant representation from all source domains. Then we use $k$ decoders to reconstruct the invariant representation for each $f_k$. TO train $E$, we use an adversarial discriminator $D'$ to enforce invariance with a min-max game, where the discriminator tries to identify the source domain of the invariant representation, and the encoder tries to fool it:

$$\min_E \max_{D'} -\sum_{i=1}^{k} \mathbb{E}_{(x,y)\sim X_k}[l(D'(E(x)), k)] \tag{1}$$

We use a gradient reversal layer $R$ to convert the min-max to single-step minimization. $R$ is the identity matrix during the forward pass, and the discriminator gradient during the backward pass, scaled by $\lambda = -1$. Then, the adversarial optimization becomes:

$$Loss = \min_{E,R(D')} -\sum_{i=1}^{k} \mathbb{E}_{(x,y)\sim X_k}[l(D'(R(E(x))), k)] \tag{2}$$

To train $D_k$, we use the masked language modeling loss from BERT and AlBERT pretraining. Parameters are initialized with AlBERT's `albert-base-uncased` weights (Lan et al., 2019).

In summary, we train a single encoder and $k$ decoders. The encoder projects our training data in the form of SentencePiece tokens (Kudo & Richardson, 2018) into a domain-invariant representation, trained with a domain discriminator. Each decoder then reconstructs the original input training tokens from the invariant representation. We use decoders because MIDAS is designed to work with our existing BERT and AlBERT classifiers, which expect SentencePiece token input. Decoders are trained with masked language modeling, where we randomly mask up to 15% of words or tokens in the input. Then, during prediction, an unlabeled, potentially drifted sample $x'$ from an unseen distribution $X'$ is converted to a domain invariant representation $E(x')$. Each decoder $D_k$ reconstructs from this invariant representation the input to its corresponding SM $f_k$.

## 4.2 RANDOMIZED LIPSCHITZ SMOOTHNESS

With a common embedding space, we can now compare relevancy of each model to an unlabeled, potentially drift sample $x'$. To present our approach, we need to introduce Lipschitz continuity.

**Definition (Lipschitz Continuity).** A function $f : \mathbb{R}^n \to \mathbb{R}^m$ is Lipschitz continuous if, for some metric space $(X, \theta)$, there exists a constant $L$ such that

$$\theta(f(x_1), f(x_2)) \leq L \cdot \theta(x_1, x_2)$$

We can extend this to define Lipschitz-smooth with respect to SM predictions using Lipschitzness from (Chen et al., 2022).

**Definition (Lipschitzness).** An SM is Lipszhitz smooth if, for some class label $C$,

$$|\Pr(f_k(x_1) = C) - Pr(f_k(x_2) = C)| \leq L_k\theta(x_1, x_2) \tag{3}$$

That is, with $L_k$ smoothness, the difference in $f_k$'s predicted labels on $x_1$ and $x_2$ is bounded by $L_k$ for all $x \in X$. However, the local value of $L_k$ can vary across the embedding space. Consequently, $f_k$ is smoother wherever $L_k$ is smaller [1]. As such, we want small $L_k$ for samples in the same class, and large $L_k$ for samples from different classes.

With these definitions, we can present our approach for finding the best-fit relevant $f_k$ for $x'$, defined as the $f_k$ with the smoothest embedding space around $x'$.

**Theorem 1.** Let the best-fit $f_k$ for a sample $x'$ be the SM that is smoothest around $x'$. We can find the best-fit $f_k$ for a particular sample $x'$, given a distance threshold $\epsilon$, by solving:

$$\arg\min_k \max_{\theta(x',x_r)\leq\epsilon|_{r=1}^N} \frac{\theta(\Pr(f_k(x')), \Pr(f_k(x_r)))}{\theta(x', x_r)} \tag{4}$$

---

[1] As $L_k \to 0$, the embedding function approaches mode collapse, where every input point is projected to the same embedding point

The $\max$ term estimates the $L_k$ value for each $f_k$ by sampling $N$ random points in an $\epsilon$-Ball around $x'$. Then, we find the $f_k$ that has the smallest $L_k$.

A key insight is that adversarial attacks exploit non-smoothness of a model's embedding space to fool classifiers, by generating a noise $\epsilon$ such that $f_k(x + \epsilon) \neq f_k(x)$. This non-smoothness occurs when $f_k$ does not capture enough training data in the region around $x$ properly; in GANs, this causes 'holes' in the latent space (Suprem et al., 2020) during image synthesis. Conversely, adversarial defenses either enforce smoothness around embedding space or on potentially perturbed inputs themselves (Das et al., 2018). Similarly, GANs can enforce 1-Lipschitzness to improve coverage of sample generation (Qin et al., 2020) .

So, given several $f_k$ with different local $L_k$ around the embedding $f_k(x + \epsilon)$, a lower $L_k$ indicates smoother embedding space *because* that SM has captured more training data in the region surrounding $x$ relative to other SMs, similar to the overlap metric calculated in (Suprem & Pu, 2022) However, even with a low $L_k$, the classification labels $y_k = f_k'(f_k(x))$, obtained from the classification module $f_k'$ of the $k$th SM, can change on perturbations around $x$. We can use probabilistic Lipschitzness to bound the probability of $y_k$ changing in the neighborhood $x + \epsilon$ as a function of the perturbation $\epsilon$.

**Definition (Probabilistic Lipschitzness).** Let $\phi : \mathbb{R} \to [0, 1]$. Given $x', x_r \sim P_X$, we say that $f_k$ is $\phi$-Lipschitz if, for all $\epsilon > 0$, there is an increasing function $\phi(\epsilon)$ such that:

$$\Pr_{x', x_r \sim P_X}[\theta(f_k(x'), f_k(x_r)) - \frac{1}{\epsilon}\theta(x', x_r) > 0] \leq \phi(\epsilon) \tag{5}$$

That is, a function that is Lipschitz by Definition 1 (L-Lipschitz) satisfies Definition 3 ($\phi$-Lipschitz) with $\phi(\epsilon) = 1$ if $\epsilon \geq 1/L$ and $\phi(\epsilon) = 0$ if $\epsilon < 1/L$.

From this, it follows that if $f_k$ satisfies the $\phi$-Lipschitz condition, then the number of samples within an $\epsilon$-Ball of $x'$ that have a different label from $x'$ is bounded by $\phi(\epsilon)$, per (Urner & Ben-David, 2013). As we move further from $x'$, the probability of a label change increases. Let $a'$ be the accuracy for $f_k$ at $x'$. If we know the accuracy drop $\alpha$ at the edge of the $\epsilon$-Ball where labels change values, we can bound the accuracy of predictions between $x'$ and some perturbed point $x_r$ in an $\epsilon$-Ball around $x'$ to:

$$a_r \geq a' - \alpha \cdot \phi(\epsilon)$$

If we calculate $\alpha$ for an SM using training examples of different labels within the margins allowed by probabilistic Lipschitzness, we find that accuracy bound depends only on choice of $\epsilon$. Consequently we can approximate $f_k$'s accuracy on $x'$ if $x_r$ is within a small $\epsilon$-Ball around $x'$ to be at least:

$$\theta(\Pr(f_k(x')), \Pr(f_k(x_r))) \geq \Pr(f_k(x') = y_k) - \alpha \cdot \phi(\epsilon) \tag{6}$$

## 4.3 PUTTING IT TOGETHER: MIDAS

Recall that the encoder projects all samples to a common invariant domain $E : \{X\}_{i=1}^k \to G$. Each SM-specific decoder $D_k$ then converts from the invariant domain to the $k$-th source domain $D_k : G \to X_k$. With this encoder-decoders framework, we can use the invariant domain $G$ as the common source domain for all SMs.

Per Definition 3, accuracy is best estimated in an $\epsilon < 1$ ball. Once $E$ is trained, we can use the class cluster centers (obtained from K-Means clustering on the embeddings) from the training data to compute a local cluster-specific $L$ for each cluster in each $f_k$. This partitioning allows us to compute local smoothness characteristics of the embedding space, simialr to the concurrent work in (Chen et al., 2022). We use the cluster centers because we want the strongest measure of smoothness for each $f_k$, and this occurs near the cluster center; for example (Suprem & Pu, 2022) and (Chen et al., 2022) both use the centroid to set up accuracy thresholds. Then, we can estimate a local $L_k$ for each $f_k$ at the cluster centers with the $m$ nearest point to the center using Equation (3). We take the maximum $L_k$ among all $f_k$ to obtain the upper bound on the local smoothness among the SMs. On this maximum, we can calculate $\epsilon = 1/\max(L_k {}_{i=1}^k)$. So, the only hyperparameter here is $m$, the number of neighbors to compute the local $L_k$ for each label cluster in each $f_k$. We explore impact of $m$ in Section 5.2.

| Dataset | Training | Testing | Oracle Acc. (Fine-Tuning) | Generalization Acc. (Held-Out) | Decrease |
|---|---|---|---|---|---|
| kaggle_short (Patel, 2021) | 31k | 9K | 0.97 | 0.57 | 42% |
| kaggle_long (Patel, 2021) | 31k | 9K | 0.98 | 0.53 | 46% |
| coaid (Cui & Lee, 2020) | 5K | 1K | 0.97 | 0.56 | 42% |
| cov19_text (Agarwal, 2020) | 2.5K | 0.5K | 0.98 | 0.58 | 41% |
| cov19_title (Agarwal, 2020) | 2.5K | 0.5K | 0.95 | 0.62 | 35% |
| rumor (Cheng et al., 2021) | 4.5K | 1K | 0.83 | 0.54 | 35% |
| cq (Mutlu et al., 2020) | 12.5K | 2K | 0.71 | 0.51 | 28% |
| miscov19 (Memon & Carley, 2020) | 4K | 0.6K | 0.68 | 0.50 | 26% |
| covid_fake (Das et al., 2021) | 4K | 2K | 0.96 | 0.52 | 45% |

**Table 1:** We use 9 covid fake news datasets to evaluate MiDAS. Here we also present a motivating experiment with respect to concept drift and generalizability: for each dataset, we train an 'oracle' on the training data. The oracle's performance on the test data is compared to an ensemble of the other 8 datasets. The latter tests the concept drift case, where models need to generalize to data distributions they have not yet encountered. There is significant accuracy drop due to concept drift, domain shift, and label overlap

Now, for an unlabeled sample $x'$, we first generate the domain invariant representation $x'_G = E(x')$. Then we perturb $x'_G$ to generate $r$ points $\{x'_G i\}_{i=1}^r$ in an $\epsilon$-Ball around $x'_G$. Using $x'_G$ and $\{x'_G i\}_{i=1}^r$, we can compute the local Lipschitz constant $L'_k$ for each $f_k$ using Equation (3). We have 2 possibilities:

1. $L'_k \geq 1/\epsilon$: $f_k$ that satisfy this condition abstain from providing predictions, since the accuracy drop is unbounded.

2. $L'_k < 1/\epsilon$: $f_k$ that satisfy this condition can provide labels, because their accuracy is bounded per Equation (6).

## 5 EVALUATION

### 5.1 EXPERIMENTAL SETUP AND DATASETS

We implemented and evaluated MIDAS on PyTorch 1.11 on a server running NVIDIA T100 GPUs. We have released our implementation code.

**MiDAS Datasets.** We use 9 fake news datasets, shown in Table 1. Where available, we used provided train-test splits; otherwise, we performed class-balanced 70:30 splits. We performed a preliminary motivating evaluation, shown in Table 1. Here, we have compared an oracle case to the concept drift case. In the oracle case, we train 9 models, one on each dataset, and then evaluate this model on its corresponding dataset's test set. This is the case where the prediction data matches the training distribution. In the concept drift case, we trained a model on 8 datasets, then evaluated on the held-out dataset. In this case, the prediction dataset, even though on the same topic of Covid-19 fake news detection, does not match the training distribution. We see significant accuracy drops, between 20% to 50%. This matches the observations in generalizability in (Suprem & Pu, 2022) and (Wahle et al., 2022).

**MiDAS Evaluation.** We evaluate MIDAS with held-one-out testing at the dataset level similar to the generalization studies approach in (Suprem & Pu, 2022). Our results are presented in Section 5.2 To test MIDAS, we first train the encoder to learn a domain invariant representation with all but one dataset. Then we evaluate MIDAS' performance on classifying the unseen, *i.e.* drifted, dataset. We repeat this for each dataset in Table 1. MIDAS' performance is compared to Snorkel (Ratner et al., 2017), EEWS (Rühling Cachay et al., 2021), an ensemble, and an 'oracle' fine-tuned AlBERT model trained on the held-out dataset.

**Ablation Study.** To further evaluate MIDAS' efficacy, we also conducted an ablation study in Section 5.3 by varying the number of training datasets, types of loss functions, masked language model trainig, and loss weights.

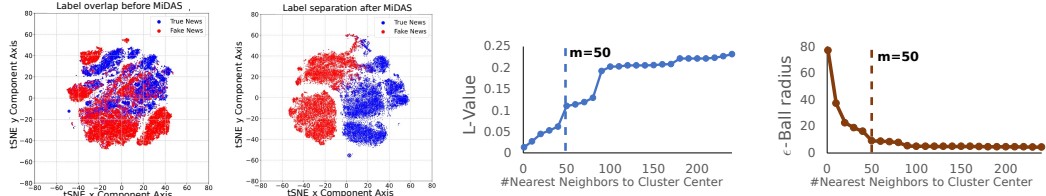

**(a)** Overlapping clusters before MiDAS    **(b)** Domain invariance after MiDAS    **(c)** L-Values for different $m$    **(d)** $\epsilon$-Ball radius for different $m$

**Figure 2: MiDAS characteristics.** Figure 2a shows tSNE projection of all 9 datasets' pretrained BERT encoder's embeddings. Figure 2b shows dataset embeddings when generated with MiDAS' encoder. The label overlaps between true news and fake news have been separated; in this case, enforcing domain invariance forces true and fake labels of all datasets to cluster. In Figure 2c and Figure 2d, we show impact of L-Value and resulting $\epsilon$ calculation for different values of $m$. We select $m = 50$ for remaining experiments.

**Adjustment of $m$.** For each experiment in Section 5.2, we sampled points in an $\epsilon$-ball around $x$, where $\epsilon$ calculated using steps Section 4.3. We explore effects of adjusting the radius of this sampling ball by changing $m$, the number of nearest neighbors, in Section 5.4. Specifically, we show that as the sampling/perturbation radius increases beyond $\epsilon$, MIDAS' accuracy decreases. Conversely, as sampling radius is reduced, MIDAS increases accuracy while sacrificing coverage.

## 5.2 MIDAS EVALUATION

We now present evaluation results for MIDAS. In each experiment, we designate a single dataset as the target dataset without labels, and the remaining datasets act as source domains. In these cases, the held-one-out dataset acts as the drifted dataset, similar to the generalization experiments in (Suprem & Pu, 2022). We follow the steps in Section 4.3 to test MIDAS, and compare classification accuracy to 5 approaches: (i) an ensemble of the training models with equal weights, (ii) a Snorkel labeler (Ratner et al., 2017) that treats each model as a labeling function, (iii) an EEWS labeler (Rühling Cachay et al., 2021) that treats each model as a labeling function, (iv) a KMP-model (Suprem & Pu, 2022) that uses KMeans clustering with proxies to compute overlap, and (v) an 'oracle' AlBERT model fine-tuned on the held-out dataset.

Figure 2 shows several characteristics of the MiDAS encoder. Figure 2b shows domain invariance in the labels. Each point is a sample from the 9 datasets; before applying MiDAS, there is significant label overlap because each dataset exists in separate domains. After applying MiDAS, datasets are projected to a domain invariance embedding. This forces sampes with the same label, irrespective of source domain, to cluster together and reduce the label overlap observed in (Suprem & Pu, 2022).

After training MIDAS' encoder, we need to compute the $\epsilon$ radius using $m$ nearest neighbors to the label cluster centers. We examine the impact of different $m$ values for computing $\epsilon$ in Figure 2c and Figure 2d. As we increase the number of neighbors used in estimating local $L$, the estimate for $L$ increases, indicating reduced smoothness the further we deviate from the cluster center. In turn, this reduces the acceptable $\epsilon$-ball radius to bound probability of label change, per (Urner & Ben-David, 2013), since $\epsilon = 1/L$. A large $m$ would significantly reduce the size of the sampling $\epsilon$-ball, and perturbations would be negligible. A small $m$ would yield a poor estimate for local $L$ and a large sampling ball. We further explore impact of changing $m$ directly on accuracy in Section 5.4. Here, we fix $m = 50$ for remaining experiments, since we observed the $\epsilon$-Ball radius generally stabilized around this value.

In Table 2, we show results of MIDAS compared to the 5 approaches discussed above. In all but one of our experiments, MIDAS outperforms other labeling schemes in classifying the unseen, drifted samples. On average, MIDAS sees a 30% increase in accuracy compared to an ensemble. Further, by using the training data itself to adaptively guide SM selection, MiDAS improves by 21% on Snorkel, 10% on EEWS and KMP.

| Dataset | Oracle Labels | No access to Labels | | | | |
|---|---|---|---|---|---|---|
| | FT-AlBert | Ensemble | Snorkel | EEWS | KMP | **MiDAS** |
| kaggle_short | 0.97 | 0.57 | 0.61 | 0.67 | 0.69 | **0.86** |
| kaggle_long | 0.98 | 0.53 | 0.63 | 0.68 | 0.70 | **0.71** |
| coaid | 0.97 | 0.56 | 0.64 | 0.74 | 0.81 | **0.84** |
| cov19_text | 0.98 | 0.58 | 0.59 | 0.68 | 0.75 | **0.79** |
| cov19_title | 0.95 | 0.62 | 0.69 | 0.75 | 0.61 | **0.81** |
| rumor | 0.83 | 0.54 | 0.59 | 0.62 | 0.45 | **0.67** |
| cq | 0.71 | 0.51 | 0.54 | 0.52 | 0.56 | **0.57** |
| miscov19 | 0.68 | 0.50 | 0.52 | **0.56** | 0.45 | 0.54 |
| covid_fake | 0.96 | 0.52 | 0.56 | 0.61 | 0.50 | **0.75** |

**Table 2:** MiDAS Evaluation: MiDAS outperforms on generalizing to unseen, drifted data points. For each held-out dataset, given 8 fine-tuned models trained on the remaining 8 datasets, MiDAS is able to select the best-fit model for each sample. Using this best-fit model, MiDAS outperforms an equal-weighted ensemble by over 30%.

| Dataset | MiDAS-Half | +Sources | +Center Loss | + Masking | +Weighted Loss |
|---|---|---|---|---|---|
| kaggle_short | 0.56 | 0.68 | 0.75 | 0.79 | 0.86 |
| kaggle_long | 0.55 | 0.62 | 0.65 | 0.68 | 0.71 |
| coaid | 0.53 | 0.71 | 0.76 | 0.79 | 0.84 |
| cov19_text | 0.54 | 0.68 | 0.72 | 0.75 | 0.79 |
| cov19_title | 0.58 | 0.71 | 0.76 | 0.78 | 0.81 |
| rumor | 0.57 | 0.61 | 0.63 | 0.65 | 0.67 |
| cq | 0.52 | 0.54 | 0.55 | 0.55 | 0.57 |
| miscov19 | 0.51 | 0.53 | 0.53 | 0.54 | 0.54 |
| covid_fake | 0.51 | 0.59 | 0.64 | 0.69 | 0.75 |

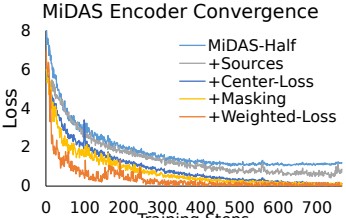

MiDAS Encoder Convergence

**Table 3: MiDAS Ablation Study.** We examined impact of different design choices here. Of note is that using masked language modeling is significantly useful in improving end-to-end accuracy. Further, adding a center loss term ensures the domain-invariance representations have separable clusters, as we see in Figure 2b. Finally, with weighted loss, we give the discriminator 2x the importance of the encoder masking loss to focus on domain invariance.

**Figure 3:** Convergence of MiDAS with each component in ablation study.

## 5.3 ABLATION STUDY

We evaluated the impact of several design and training choices for MIDAS in an ablation study in Table 3 We use a version of MiDAS trained with half of the sources with the most data points for each experiment (MiDAS-Half). This yields near-random accuracy, since this is a modified ensemble on different source datasets. Adding the remaining sources improves MiDAS' coverage and improves accuracy by 15%. We add a center loss term (He et al., 2018) to the encoder output to encourage clustering on the labels between multiple sources; increasing accuracy by 5%. Next, we added language masking to the input during the encoder-decoder training to further fine-tune the encoder for the fake-news tasks, yielding a 4% improvement. Finally, we increased the weights for the discriminator loss compared to encoder loss to emphasize domain invariance, yielding a 5% improvement for MiDAS' accuracy on fake news detection for unseen, drifted data. We compare convergence for different experiments in Figure 3: the encoder converges faster in each case. Further, adding the center and weighted losses contribute to discriminator fooling and stabilizing the discriminator loss.

## 5.4 ADJUSTMENT OF $\epsilon$

Finally, we investigate $\epsilon$ with respect to $m$, which was fixed at $m$=50. For these experiments, we investigated increasing and decreasing $m$ to, respectively, increase and decrease $\epsilon$. Increasing the neighbors increases the computed $L$, since we are using points further from the smooth cluster center. In turn, this reduces the sampling $\epsilon$-ball, so the perturbations we apply will be smaller, and in some cases, negligible. Furthermore, threshold value of $L$ needed to accept an SM's prediction is higher (since it is $1/\epsilon$), so MiDAS tolerates lower smoothness for each model, and accepts predictions from more models, resulting in higher coverage and lower overall accuracy. On the other hand, using fewer

| m-value | kaggle_short | | coaid | | rumor | | cq | |
|---|---|---|---|---|---|---|---|---|
| | F1-Score | Coverage | F1-Score | Coverage | F1-Score | Coverage | F1 | Coverage |
| m=1 | 0.97 | 0.03 | 0.97 | 0.01 | 0.86 | 0.05 | 0.81 | 0.03 |
| m=10 | 0.91 | 0.35 | 0.92 | 0.29 | 0.82 | 0.48 | 0.74 | 0.34 |
| m=20 | 0.87 | 0.65 | 0.85 | 0.73 | 0.73 | 0.63 | 0.61 | 0.59 |
| m=50 | 0.86 | 0.86 | 0.84 | 0.89 | 0.67 | 0.82 | 0.57 | 0.78 |
| m=75 | 0.73 | 0.91 | 0.78 | 0.94 | 0.62 | 0.89 | 0.53 | 0.85 |
| m=100 | 0.69 | 0.95 | 0.73 | 0.98 | 0.59 | 0.97 | 0.51 | 0.91 |
| m=150 | 0.57 | 0.99 | 0.56 | 1.00 | 0.54 | 1.00 | 0.43 | 0.95 |

**Table 4:** Impact of $m$ values: As we increase the number of nearest neighbors, we get a higher estimate for L, per Figure 2c, which reduces the $\epsilon$-Ball sampling radius and relaxes threshold for model predictions. This leads to lower accuracy with higher coverage. Decreasing $m$, in turn, increases accuracy at the cost of lower coverage.

neighbors means larger sampling ball and smaller threshold for acceptance. It is more likely for perturbed samples to be further away, yielding a higher value of L unless a corresponding model is especially smooth around that point. This would, as a result, reduce coverage, but increase accuracy.

We show this in Table 4 for several values of $m$ across 4 of our datasets. Using only the nearest neighbor yields minimal coverage. As we increase the $m$, coverage increases significantly, and accuracy approaches ensemble accuracy. Conversely, as we reduce $m$, we also reduce $L$ and consequently, the smoothness threshold to accept a prediction. This increases accuracy, since MIDAS rejects predictions that do not satisfy the threshold. However, coverage decreases as well: we show in Table 4 fewer unseen samples from the target domain can be labeled as we decrease $m$. We also see that $m$ can have outsized impact on accuracy as well: 'coaid' f1 scores drop from 0.73 to 0.56, even though coverage increases only slightly, from 0.98 to 1.0 when we increase $m$ from 100 to 150. This occurs because at $m = 150$, MiDAS' relaxed thresholds allow poorer models to provide predictions as well, reducing accuracy in the final averaged result.

## 5.5 LIMITATIONS AND FUTURE DIRECTIONS

We tested MiDAS in the scenario where fine-tuned models already exist. This constrains the MiDAS encoder, which must also train a decoder so match the inputs of the fine-tuned models that expect tokenized input. A more flexible approach would deploy models and MiDAS together, with each fine-tuned model directly trained with the MiDAS encoder. This would improve both training time, convergence, as well as accuracy, since each model would directly use the MiDAS generated encodings, instead of domain-specific reconstructions.

Furthermore, we selected $m$ using empirical observations. However, there can be technically grounded approaches as well, such as using the high-density bands from (Suprem et al., 2020; Jiang et al., 2018). In these cases, the $\alpha$-high density region of each cluster can be used to estimate a good $m$. We leave further exploration of $m$ as well as integration of fine-tuned models into the encoder training framework to future work.

## 6 CONCLUSION

We have presented MIDAS, a system for adaptively selecting best-fit model for a set of samples from drifting distributions. MIDAS uses a domain-invariance embedding to estimate local smoothness for fine-tuned models around drifting samples. By using local smoothness as a proxy for accuracy and training data relevancy, MIDAS improves on generalization accuracy across 9 fake news datasets. With MIDAS, we can detect COVID-19 related fake news with over 10% accuracy improvement over weak labeling approaches. We hope MIDAS will lead further exploration into the tradeoff between generalizability and fine-tuning, as well as research into mitigating generalization difficulties of pre-trained models.

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
