# OpenReview forum: "MiDAS: Multi-integrated Domain Adaptive Supervision for Fake News Detection"
_ICLR.cc/2023/Conference — Submitted to ICLR 2023_

### Official Review · Reviewer_WVAw · 2022-10-23

**Confidence:** 4
**Correctness:** 3
**Technical Novelty And Significance:** 2
**Empirical Novelty And Significance:** 2
**Recommendation:** 3

**Clarity, Quality, Novelty And Reproducibility:**

Clarity: The coherence of this paper can be improved. Especially, how the proposed work can solve unique challenges in fake news concept drift problems can be clarified. Notations should be described in detail.

Quality: The proposed method seems not well motivated. It is unclear how Lipschitzness smoothness is the best fit for fake news concept drift problems.

Novelty: the proposed idea is interesting. I think the intuition behind proposing this idea can be strengthened.

Reproducibility: code is provided in the supplemental material.

Minor comments: in Abstract, there is a typo: “a doman-invariant encoder”. Page 6: Ablation Study: “masked language model trainig”.


**Strength And Weaknesses:**

Strengths:

+This paper introduces the problem of concept drift in misinformation clearly. The motivation of designing a domain-adaptive approach is clear.

+The proposed framework is well described with figures and details. The first part, the encoder, aims to learn domain-invariant representations for data samples. Based on the learned representations, this paper introduces the Lipschitz continuity and extends this to find the best-fit model from multiple source models.

+Experiments are performed on multiple fake news datasets to test the performance of the proposed approach on classifying unseen/drifted distributions. Ablation studies are performed by changing the number of training datasets, types of loss functions, masked language model training, and loss weights.

Weaknesses:

First, this paper focuses on the application of fake news detection. However, the connection between the proposed model with fake news seems not strong. Very little analysis of fake news data distributions is provided. Datasets are not described in detail. The unique challenges in fake news detection are not introduced. If the proposed method is a general approach for any concept-drift problems, the experiments are only conducted on fake-news datasets.

Second, the motivation for using Lipschitz smoothness to estimate/find the best source model/function for an unlabeled and potentially drifting sample is not strong. The reasons for using this notion should be better elaborated.

Third, some notations can be better clarified. For instance, how is function theta specified is not clearly described in the methodology section. Equation 3 seems to use absolute difference, but the reason why this can be used for estimating the smoothness of source functions remains unclear. The motivation for using probabilistic Lipschitzness can be better explained. It would be more convincing if examples or proofs of changes in perturbations are provided.


**Summary Of The Paper:**

This paper presents an approach, MiDAS, to classify new samples in the situation where there are multiple domains and domain concepts drift over time. The proposed method has two components: a domain-invariant encoder, and an adapetive model selector. The encoder creates domain-invariant representations for data samples and the selector uses Lipschitz smoothness to estimate the best-fit model for a new sample. Experimental evaluation is performed on several fake-news datasets.


**Summary Of The Review:**

This paper presents a solution for classifying samples when multiple source domains are present and concept drifts exist. The target application is fake news detection, however, the connection of the proposed framework with this specific application is not strong. The motivation for using Lipschitzness should be better described.

---

> ### Author Response · Authors · 2022-11-19
> **Response to Reviewer WVAw**
>
> Thank you for the constructive comments and feedback! Our response is below.
>
> ### The unique challenges in fake news detection are not introduced.
>
> Our comment from Reviewer 1, reproduced below.
>
> Our primary motivation stems from a fine-grained generalization study we conducted across the datasets presented in the paper, in addition to the baseline accuracies in Table 1, where we tested each model on each dataset. In Table T1 below, each row is a model trained on that row’s dataset. Each column entry is the test accuracy of the model. We show that even though each dataset concerns covid fake news, classifiers trained on each dataset perform significantly worse on the related datasets, with some notable exceptions.
>
> `Table T1`
>
> | Row: Training Data | `cov_fn` | `k_short` | `coaid` | `cq`   | `k_long` | `rumor` | `c19_text` | `miscov` | `c19_title` |
> |--------------------|--------|---------|-------|------|--------|-------|----------|--------|-----------|
> | `cov_fn`             | **0.96**   | 0.43    | 0.51  | 0.44 | 0.47   | **0.75**  | 0.40      | 0.52   | **0.64**      |
> | `k_short`            | 0.52   | **0.97**    | 0.56  | 0.57 | **0.86**   | 0.52  | 0.50      | 0.48   | 0.52      |
> | `coaid`              | 0.20    | 0.53    | **0.97**  | 0.48 | 0.43   | 0.40   | **0.76**     | **0.83**   | **0.84**      |
> | `cq`                 | 0.57   | 0.57    | 0.55  | **0.54** | 0.59   | 0.42  | 0.55     | 0.41   | 0.46      |
> | `k_long`             | 0.53   | **0.71**    | 0.58  | 0.4  | **0.98**   | 0.43  | 0.54     | 0.48   | 0.54      |
> | `rumor`              | **0.67**   | 0.53    | 0.55  | 0.52 | **0.62**   | **0.83**  | 0.48     | 0.34   | **0.60**       |
> | `c19_text`           | 0.57   | **0.72**    | **0.79**  | 0.44 | **0.67**   | 0.58  | **0.98**     | 0.43   | 0.43      |
> | `miscov`             | 0.46   | 0.45    | 0.47  | 0.53 | 0.44   | 0.54  | 0.46     | **0.55**   | 0.54      |
> | `c19_title`          | 0.57   | 0.55    | **0.77**  | 0.55 | **0.62**   | **0.81**  | **0.62**     | 0.43   | **0.95**      |
>
>
> The diagonal bold represents model's accuracy on its test set and is the highest value in each row. However, models also perform well on other datasets (non-diagonal bold) when this test data that matches their training data distribution, or if they are from similar sources. This is the crux of MiDAS: automatically identifying which model best fits a specific sample by examining the neighborhood of the embedding space and model output and comparing smoothness to the training data smoothness. This also address why half-sources give poor accuracy and why increased diversity increases accuracy: we are limiting the training data space. When we increase to all data sources, the increased diversity means we have access to models that are smooth around the test sample. For example, the `coaid` model performs well on `c19_text`, `miscov`, and `c19_title`.
>
> In such cases, MiDAS excels as a weak-labeling tool. Given multiple source models, each with noisy predictions on some target domain, MiDAS can select the best-fit subset of source models to predict labels, with abstentions from remaining classifiers on a per-sample basis. Fake news is an attractive test-bed since, as we showed, models trained on datasets with similar content and task to test data have wildly varying accuracy. Out-of-distribution generalizability on such tasks is akin to weak labeling, where we have multiple noisy labels from related labeling functions.
>
> ### Second, the motivation for using Lipschitz smoothness to estimate/find the best source model/function for an unlabeled and potentially drifting sample is not strong. The reasons for using this notion should be better elaborated.
>
> Continuing from the prior response, smoothness is a useful metric to measure similarity between training and test data. It allows us to quickly determine if a classifier has a good *fit* for a specific sample, or whether it should abstain, merely by using data a classifier already has access to, namely its own embedding space and training data.
>
> ### Third, some notations can be better clarified. For instance, how is function theta specified is not clearly described in the methodology section.
>
> We have mentioned in the definition of Lipschitz continuity (Section 4.2) that theta is a distance metric to be used to compute Lipschitz score; for text embeddings, we use cosine similarity as it has shown good promise for BERT embeddings as well [1]
>
> ### Equation 3 seems to use absolute difference, but the reason why this can be used for estimating the smoothness of source functions remains unclear.
>
> In notation for euclidean norm, |a-b| is equivalent to ||a-b||.

---

> > ### Author Response · Authors · 2022-11-19
> > **Cont. Response to Reviewer WVAw**
> >
> > ### The motivation for using probabilistic Lipschitzness can be better explained. It would be more convincing if examples or proofs of changes in perturbations are provided.
> >
> > We conducted 2 additional experiments, reproduced here from other reviewer comments as well.
> >
> > #### Impact of $m$ values
> >
> > In our paper, we select $m=50$ empirically. This is based on observations of prediction coverage when varying different values of $m$.
> > That is, for different $m$ values, the coverage, or how much of the test samples MiDAS can predict confidently, changes. For low `m`, the coverage is lower since more samples do not lie close enough to any source domain. We compared an `exact` approach where MiDAS provides predictions only for points inside the lipschitz threshold, and a `fallback` approach. In the `fallback` approach we selected the smoothest model outside the threshold for each test samples MiDAS would reject with the `exact` approach. This allows us to achieve 100% coverage, where abstentions are replaced with a 'best guess' fallback approach. In this case, accuracy decreases somewhat since we are using models that do not fit the smoothness threshold; when we have low coverage to begin with, the accuracy decrease is significant. However, at higher coverage, e.g. m values of 20-50, accuracy decrease is not significant since we still use the best-fit model outside the threshold. We show below results of MiDAS tested on `kaggle_short` and `coaid`.
> >
> > `Table T2`
> >
> > | m-value | kaggle_short  |          |          | coaid    |          |          |
> > |---------|---------------|----------|----------|----------|----------|----------|
> > |         | F1-Score      | Coverage | Fallback | F1-Score | Coverage | Fallback |
> > | m=1     | 0.97          | 0.03     | 0.59     | 0.97     | 0.01     | 0.56     |
> > | m=10    | 0.91          | 0.35     | 0.64     | 0.92     | 0.29     | 0.63     |
> > | m=20    | 0.87          | 0.65     | 0.79     | 0.85     | 0.73     | 0.72     |
> > | m=50    | 0.86          | 0.86     | 0.72     | 0.84     | 0.89     | 0.81     |
> > | m=75    | 0.73          | 0.91     | 0.67     | 0.78     | 0.94     | 0.77     |
> > | m=100   | 0.69          | 0.95     | 0.62     | 0.73     | 0.98     | 0.71     |
> > | m=150   | 0.57          | 0.99     | 0.57     | 0.56     | 1.00     | 0.56     |
> >
> >
> > #### Correlation of smoothness to accuracy
> >
> > We also conducted an experiment to provide experimental justification of theoretical findings of correlation of smoothness to accuracy. We repeat the held-out cross-validation experiment in Table 2 in paper, with some changes. For each model, we do the following. First, we computed the local $L_k$ values of the training data for model $f_k$ by using the training data embeddings, and estimate $\epsilon$. Then, for each testing sample $x'$, we generate the corresponding embedding and perturb it in an $\epsilon$ ball around the sample. Then we compute the perturbation $L_{x'}$ value using Eq. 3.  If $L_{x'} > L_k$, the model abstains under MiDAS. For this experiment, We ignore abstention and compute accuracy for all samples, and compare to the ratio $L_k /L_{x'}$. Specifically, we are interested when $L_k/L_{x'} < 1$, as these indicate points that fall outside the $L_k$ smoothness threshold. This shows the correlation between smoothness and accuracy.
> >
> > `Table T3`
> >
> > | $L_k/L_{x'}$ ratio | Accuracy      |
> > |--------------------|---------------|
> > | >1                 | 0.95$\pm$0.03 |
> > | 0.9-1              | 0.93$\pm$0.04 |
> > | 0.8-0.9            | 0.87$\pm$0.04 |
> > | 0.7-0.8            | 0.82$\pm$0.05 |
> > | 0.6-0.7            | 0.79$\pm$0.05 |
> > | 0.5-0.6            | 0.72$\pm$0.06 |
> > | 0.4-0.5            | 0.69$\pm$0.06 |
> > | 0.3-0.4            | 0.65$\pm$0.12 |
> > | 0.2-0.3            | 0.61$\pm$0.11 |
> > | 0.1-0.2            | 0.58$\pm$0.15 |
> > | <0.1               | 0.55$\pm$0.09 |
> >
> > As the sample becomes less smooth than the baseline $L_k$, accuracy decreases and accuracy variance increases, yielding more erratic performance. Closer to the model smoothness, accuracy is higher and more stable.
> >
> >
> > ### Quality: The proposed method seems not well motivated. It is unclear how Lipschitzness smoothness is the best fit for fake news concept drift problems.
> >
> > Please see response part 1 and 2
> >
> >
> >
> > [1] Coenen et. al. (2019). Visualizing and Measuring the Geometry of BERT. NeurIPS.

---

### Official Review · Reviewer_i2cY · 2022-10-24

**Confidence:** 4
**Correctness:** 3
**Technical Novelty And Significance:** 3
**Empirical Novelty And Significance:** 3
**Recommendation:** 5

**Clarity, Quality, Novelty And Reproducibility:**

The paper is well-written and easy to follow.

Typos:
section 4.1:  "TO train E" -> "To train E"
caption of Figure 1:  should be  {M}_{(i = 1)}^k

Given the prior work local Lipschitz smoothness (Chen et al., 2022), the novelty of this paper is thin. It'd be great if the proposed method could be compared with the method LIGER from (Chen et al., 2022).

Any t-test results for the comparisons in Table 2 and 3?

**Strength And Weaknesses:**

Strength:

1. This paper focuses on an interesting problem, the detection of fake news, especially Covid-19 news. It is impactful and has a sociological significance.

2. The idea of extending local Lipschitz smoothness in measuring the relevance of a new sample to existing models is somewhat novel.

3. The paper is illustrative with an ablation study by varying the number of training datasets, types of loss functions, masked language model training, and loss weights.

Weakness:

1. The main idea of this paper is based on prior work: local Lipschitz smoothness from (Chen et al., 2022). However, comparison results with the method  LIGER from (Chen et al., 2022) are missing.

2. The sizes of the datasets used in the evaluation are quite small and the numbers of samples are in the thousands. To demonstrate the scalability of the proposed method, much larger datasets are recommended.


**Summary Of The Paper:**

The authors in this paper present Multi-Integrated Domain Adaptive Supervision (MiDAS), a framework for fake news detection that ranks the relevance of existing models to new samples. MiDAS learns domain-invariant representations by integrating multiple pre-trained and fine-tuned models with their training data. To determine each model's relevance to a new sample, MiDAS applies local Lipschitz smoothness of the invariant representation in latent space. Empirical results show MIDAS improves significantly on generalization accuracy across 9 diverse fake news datasets.

**Summary Of The Review:**

This paper presents a multi-domain adaptative approach for early fake news detection, which integrates pre-trained and fine-tuned models along with their training data to learn a domain-invariant representation. In this representation, the proposed framework adaptively selects the highest-ranked model to perform classification according to the relevancy between a new sample and model training datasets.

In general, the idea of using randomized Lipschitz smoothness measure to generate source model relevancy rankings is reasonable and effective for fake news detection.

However, the contribution of the paper is incremental given the prior work local Lipschitz smoothness (Chen et al., 2022).

There are missing technical details:
1. What are the t-test results for Table 2 and 3?
2. What are the results in terms of other metrics such as F1 and coverage for Table 2 and 3?
3. Complexity analysis of the proposed method?

To demonstrate the scalability and efficiency of the proposed method, evaluation on much larger datasets is recommended.

---

> ### Author Response · Authors · 2022-11-19
> **Response to Reviewer i2cY**
>
> Thank you for the constructive comments and feedback! Our response is below.
>
>
> ### The main idea of this paper is based on prior work
>
> While the work in Chen et al. (2022) could be considered prior work, we believe we arrived at these approaches independently. As such, we have referenced their work to show some overlap. However, there are some key differences between our approaches: we apply a randomized Lipschitzness test; instead of KMeans directly on the embeddings with C classes, where C = number of prediction classes, we use multiple proxies for each class. Since the embeddings are not globally smooth but locally smooth, we cluster into C*p clusters, where p is a constant, e.g. 10; this value can be determined with ELBOW or any off-the-shelf optimal cluster mechanism. So, instead of 2 clusters for fake/true detection, we would use 20 clusters during the clustering step, and compute a local L threshold for each cluster. Finally, we have focused MiDAS on fake news detection as highly relevant to current research. As such, we had not provided a comparison to Chen's approach. We performed a preliminary comparison on the text-datasets in Chen, and show competitive results; ours are slightly better in some cases because we used randomized Lipschitz and multiple proxies for each classification label as clusters. We used BERT as our foundation model.
>
> `Table T1`
>
> | Dataset         |  Chen (2022) |    MiDAS |
> |-----------------|--------------|----------|
> | Spam (Youtube)  |         0.95 |     0.96 |
> | Weather         |         0.98 |     0.98 |
> | Spouse          |         0.52 |     0.55 |
>
>
> ### The sizes of the datasets used in the evaluation are quite small and the numbers of samples are in the thousands. To demonstrate the scalability of the proposed method, much larger datasets are recommended.
>
> We extended Table T1 above to include comparisons to other recent works on common WS datasets. We include in our revision these results in an additional section detailing generalizability of our approach, while maintaining the initial exploration of the idea with fake news as a natural example of data where different datasets have similar task, but different distributions.
>
> `Table T2`
>
> | Dataset            |  Chen (2022) |    FS[1] | WeaSEL[2] |    MiDAS |
> |--------------------|--------------|----------|-----------|----------|
> | Spam (YouTube)     |         0.95 |     0.92 |     -     |     0.96 |
> | Weather            |         0.98 |     0.88 |     -     |     0.98 |
> | Spouse             |         0.52 |     0.50 |     0.52  |     0.55 |
> | Yelp               |         -    |     -    |     -     |     0.95 |
> | IMDB               |         -    |     -    |     0.82  |     0.88 |
>
>
> ### Typos: section 4.1: "TO train E" -> "To train E" caption of Figure 1: should be {M}_{(i = 1)}^k
>
> We have fixed these and others mentioned by reviewers.
>
> ### Given the prior work local Lipschitz smoothness (Chen et al., 2022), the novelty of this paper is thin. It'd be great if the proposed method could be compared with the method LIGER from (Chen et al., 2022).
>
> Please see response part 1,2
>
> ### Any t-test results for the comparisons in Table 2 and 3?
>
> We took the mean of multiple runs for MiDAS and Ensemble for Table 2 in paper, provided below with standard deviation (along with FT-BERT, instead of FT-AlBERT oracle results).
>
> | Dataset       |     FT-BERT     |      Ensemble   |      MiDAS      |
> |---------------|-----------------|-----------------|-----------------|
> | kaggle_short  |   0.97$\pm$0.01 |   0.57$\pm$0.02 |   0.86$\pm$0.02 |
> | kaggle_long   |   0.98$\pm$0.01 |   0.53$\pm$0.02 |   0.71$\pm$0.03 |
> | coaid         |   0.97$\pm$0.02 |   0.56$\pm$0.03 |   0.84$\pm$0.03 |
> | cov19_text    |   0.98$\pm$0.01 |   0.58$\pm$0.02 |   0.79$\pm$0.02 |
> | cov19_title   |   0.95$\pm$0.01 |   0.62$\pm$0.02 |   0.81$\pm$0.03 |
> | rumor         |   0.83$\pm$0.02 |   0.54$\pm$0.03 |   0.67$\pm$0.04 |
> | cq            |   0.71$\pm$0.01 |   0.51$\pm$0.03 |   0.57$\pm$0.04 |
> | miscov19      |   0.68$\pm$0.02 |   0.50$\pm$0.02 |   0.54$\pm$0.02 |
> | covid_fake    |   0.96$\pm$0.01 |   0.52$\pm$0.02 |   0.75$\pm$0.03 |
>
> We can, in our revision, also provide the standard deviation for Table 3 in paper.

---

> > ### Author Response · Authors · 2022-11-19
> > **Cont. Response to Reviewer i2cY**
> >
> > ### What are the results in terms of other metrics such as F1 and coverage for Table 2 and 3?
> >
> > We reproduce Table T1 from R1 (Reviewer 35Bs), where we further discussed F-score and coverage on 2 datasets.
> >
> > In our paper, we select $m=50$ empirically. This is based on observations of prediction coverage when varying different values of $m$.
> > That is, for different $m$ values, the coverage, or how much of the test samples MiDAS can predict confidently, changes. For low `m`, the coverage is lower since more samples do not lie close enough to any source domain. We compared an `exact` approach where MiDAS provides predictions only for points inside the lipschitz threshold, and a `fallback` approach. In the `fallback` approach we selected the smoothest model outside the threshold for each test samples MiDAS would reject with the `exact` approach. This allows us to achieve 100% coverage, where abstentions are replaced with a 'best guess' fallback approach. In this case, accuracy decreases somewhat since we are using models that do not fit the smoothness threshold; when we have low coverage to begin with, the accuracy decrease is significant. However, at higher coverage, e.g. m values of 20-50, accuracy decrease is not significant since we still use the best-fit model outside the threshold. We show below results of MiDAS tested on `kaggle_short` and `coaid`.
> >
> > `Table T1`
> >
> > | m-value | kaggle_short  |          |          | coaid    |          |          |
> > |---------|---------------|----------|----------|----------|----------|----------|
> > |         | F1-Score      | Coverage | Fallback | F1-Score | Coverage | Fallback |
> > | m=1     | 0.97          | 0.03     | 0.59     | 0.97     | 0.01     | 0.56     |
> > | m=10    | 0.91          | 0.35     | 0.64     | 0.92     | 0.29     | 0.63     |
> > | m=20    | 0.87          | 0.65     | 0.79     | 0.85     | 0.73     | 0.72     |
> > | m=50    | 0.86          | 0.86     | 0.72     | 0.84     | 0.89     | 0.81     |
> > | m=75    | 0.73          | 0.91     | 0.67     | 0.78     | 0.94     | 0.77     |
> > | m=100   | 0.69          | 0.95     | 0.62     | 0.73     | 0.98     | 0.71     |
> > | m=150   | 0.57          | 0.99     | 0.57     | 0.56     | 1.00     | 0.56     |
> >
> > ### Complexity analysis of the proposed method?
> >
> > Complexity of computing L-scores is $O(n)$: Given a trained model backbone $f_i$, training data $X_i$ with $N$ samples (and their respective embeddings $E_i = f_i(X_i)$), clusters on $X_i$ with their centers indexed with a KD_Tree, we perform the following for each sample to compute the L-thresholds:
> >
> > 1. Query KD tree for corresponding training cluster($O(\log p)$ for $p$ clusters, $p<<N$, so ammortized to constant time)
> > 2. Compute lipschitz score with respect to cluster center using Eq 3. ($O(1)$)
> >
> > Finally, we take the max for the L-scores of each cluster to obtaint he threshold ($O(n)$). Total runtime, simplified: $O(n)$
> >
> > Runtime for using Lipschitz scores: $O(mn)$, where $m$ is the number of neighbors (See Table 4 in paper). If $m$ is constant, as in our approach, then this can be simplified to $O(n)$ as well.
> >
> > ### To demonstrate the scalability and efficiency of the proposed method, evaluation on much larger datasets is recommended.
> >
> > Please see response part 1,2
> >
> >
> > [1] Fu et. al. (2020). Fast and three-rious: Speeding up weak supervision with triplet methods. ICML.
> > [2] Ruhling et. al. (2021). End-to-end weak supervision. NeurIPS.

---

### Official Review · Reviewer_5EyG · 2022-10-24

**Confidence:** 2
**Correctness:** 3
**Technical Novelty And Significance:** 3
**Empirical Novelty And Significance:** 2
**Recommendation:** 6

**Clarity, Quality, Novelty And Reproducibility:**

**Clarity**

Given that the paper introduces complex ideas, I'd suggest providing more details about the  approach to solution as well as the proposed method in a few sections. Some material may0Car be

Introduction

"This degree of knowledge is defined by the overlap between an unlabeled sample and existing models’ training datasets"  - overlap in what sense?

3 PROBLEM SETUP AND STRATEGY

"... where we have access to the training data Xi and weights wi." - Could you provide a more elaborated definition on "w_i"?  weights as model parameters?

"Each SM yields hidden embeddings through a feature extractor backbone, or foundation model" - this implies certain assumpton sabout the Source Models. This should be stated clearly in the scope fo this work.


4.

"we use the masked language modeling loss from BERT and AlBERT pretraining."  - please, explain the choice of the LMs


4.2 RANDOMIZED LIPSCHITZ SMOOTHNESS - I'd suggest moving this section partially or entirely to apendix to provide more space for explaning some MiDAS specific details as well as other details o fthe current work to increase clarity.

5.

"NVIDIA T100 GPUs."  - I was not able to verify Nvidia T100 GPUs (there are V100 and T1000 or T400). Either there's a typo, or  please provide a footnote with more details if it's a speacial case.


"an ‘oracle’ fine-tuned AlBERT model trained on the held-out dataset." - if AlBERT is used for the oracle, is it BERT's or AlBERT's LMs that are used fro MiDAS.


**Novelty**

From what I could judge by related work section and the refernces, the paper introduces a novel and original approach.



**Strength And Weaknesses:**

strengths
* the paper proposes a novel approach  that is superior to the presneted previosu approaches
* the paper provides concise overview of the Lipshitz smoothness concept to explain its efficiency for the problem
* the analysis is conducted on multiple test sets
* ablation study and parameter analysis provides additional soundness to the paper


weaknesses:
* more as a point of observation, the propsed method  does not use any features that makes it specifivcally focused on fake news detection or COVID dataset. It could have been applied and verified on groups of datasets for a variety of classificationn tasks. That could have made the work more impactful, and the empirical groundings more sound. This is my major concern towards this work: if the porposed method is correct, it should generalize across any text classification tasks (that satisfy Lipschitz smoothness requirements) and applying it towards a certain subrgroup of text classification (even as "trendy" as fake news and covid fake news in particular) diminishes the potential contribution, making the paper more apt for a focused workshop rather than the main conference venue.

* there's a few novel points introduced in the paper: a use of certain architecture, application of concept of Lipschitz smoothness, parameter analysis. However, there's some lack of clarity. The paper would benefit from more detailed explaining  why certain choices were made, what is the intended scope of the method, does it imply any retrsictions on the data or the models involved, etc?

**Summary Of The Paper:**

The paper introduces a novel approach to  the problem of out-of-domain  texts for fake news classification.  The method involves an adaptive model selector which employs Lipschitz smoothness concept to estimate model's relevancy.
The paper shows superiority over a number of baselines and previously introduced methods.



**Summary Of The Review:**

The main observation about this paper is that it seem to introduce a general method applicable to text classification, yet only verifies it for quite a narrow, albeit important, area of covid-related fake news detection.

---

> ### Author Response · Authors · 2022-11-19
> **Response to Reviewer 5EyG**
>
> Thank you for the constructive comments and feedback! Our response is below.
>
> ### Additional general-purpose experiments
>
> We performed additional experiments on common weak-supervision tasks, and have provided these results below; we can include these results in an additional section detailing generalizability of our approach, while maintaining the initial exploration of the idea with fake news.
>
> `Table T1`
>
> | Dataset            |  Chen (2022) |    FS[1] | WeaSEL[2] |    MiDAS |
> |--------------------|--------------|----------|-----------|----------|
> | Spam (YouTube)     |         0.95 |     0.92 |     -     |     0.96 |
> | Weather            |         0.98 |     0.88 |     -     |     0.98 |
> | Spouse             |         0.52 |     0.50 |     0.52  |     0.55 |
> | Yelp               |         -    |     -    |     -     |     0.95 |
> | IMDB               |         -    |     -    |     0.82  |     0.88 |
>
>
> ### The paper would benefit from more detailed explaining why certain choices were made, what is the intended scope of the method, does it imply any retrsictions on the data or the models involved, etc?
>
> We believe we have answered these concerns with our rebuttal to R1, reproduced here.
>
> --
>
> We had performed additional experiments to support our design choices; our experiments explored impact of $m$ values, $L_k$ smoothness, and diversity of the source data. We have provided some results here, and can select the best-fit results for a final revision.
>
>
> #### Impact of $m$ values
>
> In our paper, we select $m=50$ empirically. This is based on observations of prediction coverage when varying different values of $m$.
> That is, for different $m$ values, the coverage, or how much of the test samples MiDAS can predict confidently, changes. For low `m`, the coverage is lower since more samples do not lie close enough to any source domain. We compared an `exact` approach where MiDAS provides predictions only for points inside the lipschitz threshold, and a `fallback` approach. In the `fallback` approach we selected the smoothest model outside the threshold for each test samples MiDAS would reject with the `exact` approach. This allows us to achieve 100% coverage, where abstentions are replaced with a 'best guess' fallback approach. In this case, accuracy decreases somewhat since we are using models that do not fit the smoothness threshold; when we have low coverage to begin with, the accuracy decrease is significant. However, at higher coverage, e.g. m values of 20-50, accuracy decrease is not significant since we still use the best-fit model outside the threshold. We show below results of MiDAS tested on `kaggle_short` and `coaid`.
>
> Table R1
>
> | m-value | kaggle_short  |          |          | coaid    |          |          |
> |---------|---------------|----------|----------|----------|----------|----------|
> |         | F1-Score      | Coverage | Fallback | F1-Score | Coverage | Fallback |
> | m=1     | 0.97          | 0.03     | 0.59     | 0.97     | 0.01     | 0.56     |
> | m=10    | 0.91          | 0.35     | 0.64     | 0.92     | 0.29     | 0.63     |
> | m=20    | 0.87          | 0.65     | 0.79     | 0.85     | 0.73     | 0.72     |
> | m=50    | 0.86          | 0.86     | 0.72     | 0.84     | 0.89     | 0.81     |
> | m=75    | 0.73          | 0.91     | 0.67     | 0.78     | 0.94     | 0.77     |
> | m=100   | 0.69          | 0.95     | 0.62     | 0.73     | 0.98     | 0.71     |
> | m=150   | 0.57          | 0.99     | 0.57     | 0.56     | 1.00     | 0.56     |
>
>
> Response continued

---

> > ### Author Response · Authors · 2022-11-19
> > **Cont. Response to Reviewer 5EyG**
> >
> > #### Correlation of smoothness to accuracy
> >
> > We also conducted an experiment to provide experimental justification of theoretical findings of correlation of smoothness to accuracy. We repeat the held-out cross-validation experiment in Table 2, with some changes. For each model, we do the following. First, we computed the local $L_k$ values of the training data for model $f_k$ by using the training data embeddings, and estimate $\epsilon$. Then, for each testing sample $x'$, we generate the corresponding embedding and perturb it in an $\epsilon$ ball around the sample. Then we compute the perturbation $L_{x'}$ value using Eq. 3.  If $L_{x'} > L_k$, the model abstains under MiDAS. For this experiment, We ignore abstention and compute accuracy for all samples, and compare to the ratio $L_k /L_{x'}$. Specifically, we are interested when $L_k/L_{x'} < 1$, as these indicate points that fall outside the $L_k$ smoothness threshold. This shows the correlation between smoothness and accuracy.
> >
> > Table R2
> >
> > | $L_k/L_{x'}$ ratio | Accuracy      |
> > |--------------------|---------------|
> > | >1                 | 0.95$\pm$0.03 |
> > | 0.9-1              | 0.93$\pm$0.04 |
> > | 0.8-0.9            | 0.87$\pm$0.04 |
> > | 0.7-0.8            | 0.82$\pm$0.05 |
> > | 0.6-0.7            | 0.79$\pm$0.05 |
> > | 0.5-0.6            | 0.72$\pm$0.06 |
> > | 0.4-0.5            | 0.69$\pm$0.06 |
> > | 0.3-0.4            | 0.65$\pm$0.12 |
> > | 0.2-0.3            | 0.61$\pm$0.11 |
> > | 0.1-0.2            | 0.58$\pm$0.15 |
> > | <0.1               | 0.55$\pm$0.09 |
> >
> > As the sample becomes less smooth than the baseline $L_k$, accuracy decreases and accuracy variance increases, yielding more erratic performance. Closer to the model smoothness, accuracy is higher and more stable.
> >
> > #### Generalization study
> >
> > We also conducted fine-grained generalization study in addition to Table 1, where we tested each model on each dataset. Each row is a model trained on that row’s dataset. Each column entry is the test accuracy of the model. We show that even though each dataset concerns covid fake news, classifiers trained on each dataset perform significantly worse on the related datasets, with some notable exceptions.
> >
> > Table R3
> >
> > | Row: Training Data | `cov_fn` | `k_short` | `coaid` | `cq`   | `k_long` | `rumor` | `c19_text` | `miscov` | `c19_title` |
> > |--------------------|--------|---------|-------|------|--------|-------|----------|--------|-----------|
> > | `cov_fn`             | **0.96**   | 0.43    | 0.51  | 0.44 | 0.47   | **0.75**  | 0.40      | 0.52   | **0.64**      |
> > | `k_short`            | 0.52   | **0.97**    | 0.56  | 0.57 | **0.86**   | 0.52  | 0.50      | 0.48   | 0.52      |
> > | `coaid`              | 0.20    | 0.53    | **0.97**  | 0.48 | 0.43   | 0.40   | **0.76**     | **0.83**   | **0.84**      |
> > | `cq`                 | 0.57   | 0.57    | 0.55  | **0.54** | 0.59   | 0.42  | 0.55     | 0.41   | 0.46      |
> > | `k_long`             | 0.53   | **0.71**    | 0.58  | 0.4  | **0.98**   | 0.43  | 0.54     | 0.48   | 0.54      |
> > | `rumor`              | **0.67**   | 0.53    | 0.55  | 0.52 | **0.62**   | **0.83**  | 0.48     | 0.34   | **0.60**       |
> > | `c19_text`           | 0.57   | **0.72**    | **0.79**  | 0.44 | **0.67**   | 0.58  | **0.98**     | 0.43   | 0.43      |
> > | `miscov`             | 0.46   | 0.45    | 0.47  | 0.53 | 0.44   | 0.54  | 0.46     | **0.55**   | 0.54      |
> > | `c19_title`          | 0.57   | 0.55    | **0.77**  | 0.55 | **0.62**   | **0.81**  | **0.62**     | 0.43   | **0.95**      |
> >
> >
> > The diagonal bold represents model's accuracy on its test set and is the highest value in each row. However, models also perform well on other datasets (non-diagonal bold) when this test data that matches their training data distribution, or if they are from similar sources. This is the crux of MiDAS: automatically identifying which model best fits a specific sample by examining the neighborhood of the embedding space and model output and comparing smoothness to the training data smoothness. This also address why half-sources give poor accuracy and why increased diversity increases accuracy: we are limiting the training data space. When we increase to all data sources, the increased diversity means we have access to models that are smooth around the test sample. For example, the `coaid` model performs well on `c19_text`, `miscov`, and `c19_title`.
> >
> >
> > Response continued

---

> > > ### Author Response · Authors · 2022-11-19
> > > **Cont. Response to Reviewer 5EyG**
> > >
> > > #### The intended scope of the method; Does it imply any restrictions on the data or the models involved
> > >
> > > We believe MiDAS can be used as a weak-labeling tool. Given multiple source models, each with noisy predictions on some target domain, MiDAS can select the best-fit subset of source models to predict labels, with abstentions from remaining classifiers on a per-sample basis. Labels generated through MiDAS are less noisy, as we showed in Table R2 above: with MiDAS, using only labels inside the smoothness threshold, we have minimal variance in accuracy, whereas relying on the model for all samples without considering local smoothness yields lower *and* noisier accuracy.
> > >
> > > With respect to restrictions on data or models, we presented one such restriction in the feature extractor backbone, where we have mentioned we used foundation models due to their smoother embeddings. We can relax this constraint by requiring instead that MiDAS-compatible classifiers be trained with a smoothness enforcing loss on the latent representation, e.g. GANs with 1-Lipschitzness.
> > >
> > > --
> > >
> > >
> > > ### "This degree of knowledge is defined by the overlap between an unlabeled sample and existing models’ training datasets" - overlap in what sense?
> > >
> > > The reference (Suprem and Pu, 2022) re-introduces a cluster overlap metric (point-proximity) used in ecology for dataset overlap computation. Here, overlap is computed as the fraction of samples in a cluster A that are closer to samples in cluster B than samples in cluster A. a higher value indicates higher overlap; as overlap -> 1, this means A is fully contained in B. The choice of distance metric is important; we would use cosine similarity on BERT embeddings when computing overlap, due to the exploration of BERT embeddings geometry in [3].
> > >
> > > ### "... where we have access to the training data Xi and weights wi." - Could you provide a more elaborated definition on "w_i"? weights as model parameters?
> > >
> > > These are parameters of our model. When we mention ${f_i}_{i=1}^k$, this indicates $k$ models, each model with index $f_i$. For each model, we have access to its corresponding training data $X_i$ (where there are $k$ datasets, one for each $f_i$), as well as the parameters of the model $w_i$ (where, again, we have $k$ sets of parameters).
> > >
> > >
> > > ### "Each SM yields hidden embeddings through a feature extractor backbone, or foundation model" - this implies certain assumpton sabout the Source Models. This should be stated clearly in the scope fo this work.
> > >
> > > We have added the following constraints to clarify our statement (also mentioned for R1):
> > >
> > > Each SM yields hidden embeddings through a feature extractor backbone, or foundation model (Bommasani et al., 2021). The foundation model used for a task should intuitively correspond to the modality of the task; for example with fake news classification, we use pre-trained BERT and AlBERT transformers as our foundation models. Similarly, for image classification, ResNet or ViT [4] may be preferable.
> > >
> > >
> > > ### "we use the masked language modeling loss from BERT and AlBERT pretraining." - please, explain the choice of the LMs
> > >
> > > We used AlBERT initially for its smaller model size; it allowed us to iterate faster on preliminary experimental results before moving to BERT. We stayed with BERT due to our familairy with the architecture and superior performance since [5] notes that CLIP performs worse than BERT/RoBERTa counterparts on text encoding.
> > >
> > > ### 4.2 RANDOMIZED LIPSCHITZ SMOOTHNESS - I'd suggest moving this section partially or entirely to apendix to provide more space for explaning some MiDAS specific details as well as other details o fthe current work to increase clarity.
> > >
> > > We can make this change to our revision, along with adding results on more general datasets (as in Table R1 above)
> > >
> > >
> > > ### "NVIDIA T100 GPUs." - I was not able to verify Nvidia T100 GPUs (there are V100 and T1000 or T400). Either there's a typo, or please provide a footnote with more details if it's a speacial case.
> > >
> > > This is a typo. Our initial experiments were done with T4 GPUs, and later with V100 GPUs. We mistyped T4->V100 as T100.
> > >
> > > ## "an ‘oracle’ fine-tuned AlBERT model trained on the held-out dataset." - if AlBERT is used for the oracle, is it BERT's or AlBERT's LMs that are used for MiDAS.
> > >
> > > We used BERT for MiDAS. Differences in oracle between AlBERT and BERT were negligible, but our revision replaces the oracle AlBERT accuracies with BERT accuracies.
> > >
> > > [1] Fu et. al. (2020). Fast and three-rious: Speeding up weak supervision with triplet methods. ICML.
> > > [2] Ruhling et. al. (2021). End-to-end weak supervision. NeurIPS.
> > > [3] Coenen et. al. (2019). Visualizing and Measuring the Geometry of BERT. NeurIPS.
> > > [4] Dosovitskiy, Alexey, et al. (2020) An image is worth 16x16 words: Transformers for image recognition at scale.
> > > [5] Shen et. al. (2022). How much can CLIP benefit Vision-and-Language Tasks? ICLR.

---

### Official Review · Reviewer_35Bs · 2022-10-25

**Confidence:** 3
**Correctness:** 3
**Technical Novelty And Significance:** 3
**Empirical Novelty And Significance:** 3
**Recommendation:** 5

**Clarity, Quality, Novelty And Reproducibility:**

Given the prior work on Lipschitz smoothness by Chen et al., (2022), the novelty of this paper is very small. It will be great if the author could compared their work with the work from the reference above.
The author should look through the problem setup and strategy (Section 3):
"... where we have access to the training data Xi and weights wi." - Could you provide a more elaborated definition on "w_i"? Using weights as model parameters? "Each SM yields hidden embeddings through a feature extractor backbone or foundation model"—this implies certain assumptions about the source models. This should be stated clearly in the scope of this work.-- please check~

**Details Of Ethics Concerns:**

No comments

**Strength And Weaknesses:**

Strength:
1. This paper focuses on an exciting problem, detecting fake news, especially COVID-19 news. It is impactful and has sociological value.
2. Using local Lipschitz smoothness is a new way to figure out how well a new sample fits existing models, which is very exciting.
3. An ablation study is used as an example in the study. The number of training datasets, types of loss functions, masked language model training, and loss weights are all changed.

Weakness:
1.
2.
3.

**Summary Of The Paper:**

The paper introduces MiDAS, a novel method to address the problem of out-of-domain texts for fake news classification. MiDAS applies an adaptive model selector that utilizes the Lipschitz smoothness idea to estimate the model's relevancy. The author shows supremacy over several baselines and previously introduced methods.

**Summary Of The Review:**

In order to learn a domain-invariant representations, the multi-domain adaptive method for premature false news detection presented in this study combines pre-trained and improved models with their training examples. According to the similarities between a new sample and model training datasets, the designed system adaptively selects the highest-ranked model to carry out classification in this representation. In general, it makes perfect sense and performs well for detecting fake news to use randomized Lipschitz smoothness metric to produce source model relevancy rankings.

A few novel points are introduced in the paper: the use of particular architecture, the application of the concept of Lipschitz smoothness, and parameter analysis. However, there is some lack of clarity. The paper would benefit from a more detailed explanation of why confident choices were made, the intended scope of the method, does it imply any restrictions on the data or the models involved, etc.

---

> ### Author Response · Authors · 2022-11-19
> **Response to Reviewer 35Bs**
>
> Thank you for the constructive comments and feedback! Our response is below.
>
> ### Could you provide a more elaborated definition on "w_i"?
>
> These are parameters of our model. When we mention ${f_i}_{i=1}^k$, this indicates $k$ models, each model with index $f_i$. For each model, we have access to its corresponding training data $X_i$ (where there are $k$ datasets, one for each $f_i$), as well as the parameters of the model $w_i$ (where, again, we have $k$ sets of parameters).
>
>
> ### "Each SM yields hidden embeddings through a feature extractor backbone or foundation model"—this implies certain assumptions about the source models
>
> We have added the following constraints to clarify our statement:
>
> Each SM yields hidden embeddings through a feature extractor backbone, or foundation model (Bommasani et al., 2021). The foundation model used for a task should intuitively correspond to the modality of the task; for example with fake news classification, we use pre-trained BERT and AlBERT transformers as our foundation models. Similarly, for image classification, ResNet [1] or ViT [2] may be preferable.
>
> ### The paper would benefit from a more detailed explanation of why confident choices were made
>
> We had performed additional experiments to support our design choices; our experiments explored impact of $m$ values, $L_k$ smoothness, and diversity of the source data. We have provided some results here, and can select the best-fit results for a final revision.
>
>
> #### Impact of $m$ values
>
> In our paper, we select $m=50$ empirically. This is based on observations of prediction coverage when varying different values of $m$.
> That is, for different $m$ values, the coverage, or how much of the test samples MiDAS can predict confidently, changes. For low `m`, the coverage is lower since more samples do not lie close enough to any source domain. We compared an `exact` approach where MiDAS provides predictions only for points inside the lipschitz threshold, and a `fallback` approach. In the `fallback` approach we selected the smoothest model outside the threshold for each test samples MiDAS would reject with the `exact` approach. This allows us to achieve 100% coverage, where abstentions are replaced with a 'best guess' fallback approach. In this case, accuracy decreases somewhat since we are using models that do not fit the smoothness threshold; when we have low coverage to begin with, the accuracy decrease is significant. However, at higher coverage, e.g. m values of 20-50, accuracy decrease is not significant since we still use the best-fit model outside the threshold. We show below results of MiDAS tested on `kaggle_short` and `coaid`.
>
> `Table T1`
>
> | m-value | kaggle_short  |          |          | coaid    |          |          |
> |---------|---------------|----------|----------|----------|----------|----------|
> |         | F1-Score      | Coverage | Fallback | F1-Score | Coverage | Fallback |
> | m=1     | 0.97          | 0.03     | 0.59     | 0.97     | 0.01     | 0.56     |
> | m=10    | 0.91          | 0.35     | 0.64     | 0.92     | 0.29     | 0.63     |
> | m=20    | 0.87          | 0.65     | 0.79     | 0.85     | 0.73     | 0.72     |
> | m=50    | 0.86          | 0.86     | 0.72     | 0.84     | 0.89     | 0.81     |
> | m=75    | 0.73          | 0.91     | 0.67     | 0.78     | 0.94     | 0.77     |
> | m=100   | 0.69          | 0.95     | 0.62     | 0.73     | 0.98     | 0.71     |
> | m=150   | 0.57          | 0.99     | 0.57     | 0.56     | 1.00     | 0.56     |
>
>
> Response continued

---

> > ### Author Response · Authors · 2022-11-19
> > **Cont. Response to Reviewer 35Bs**
> >
> > #### Correlation of smoothness to accuracy
> >
> > We also conducted an experiment to provide experimental justification of theoretical findings of correlation of smoothness to accuracy. We repeat the held-out cross-validation experiment in Table 2 in paper, with some changes. For each model, we do the following. First, we computed the local $L_k$ values of the training data for model $f_k$ by using the training data embeddings, and estimate $\epsilon$. Then, for each testing sample $x'$, we generate the corresponding embedding and perturb it in an $\epsilon$ ball around the sample. Then we compute the perturbation $L_{x'}$ value using Eq. 3.  If $L_{x'} > L_k$, the model abstains under MiDAS. For this experiment, We ignore abstention and compute accuracy for all samples, and compare to the ratio $L_k /L_{x'}$. Specifically, we are interested when $L_k/L_{x'} < 1$, as these indicate points that fall outside the $L_k$ smoothness threshold. This shows the correlation between smoothness and accuracy.
> >
> > `Table T2`
> >
> > | $L_k/L_{x'}$ ratio | Accuracy      |
> > |--------------------|---------------|
> > | >1                 | 0.95$\pm$0.03 |
> > | 0.9-1              | 0.93$\pm$0.04 |
> > | 0.8-0.9            | 0.87$\pm$0.04 |
> > | 0.7-0.8            | 0.82$\pm$0.05 |
> > | 0.6-0.7            | 0.79$\pm$0.05 |
> > | 0.5-0.6            | 0.72$\pm$0.06 |
> > | 0.4-0.5            | 0.69$\pm$0.06 |
> > | 0.3-0.4            | 0.65$\pm$0.12 |
> > | 0.2-0.3            | 0.61$\pm$0.11 |
> > | 0.1-0.2            | 0.58$\pm$0.15 |
> > | <0.1               | 0.55$\pm$0.09 |
> >
> > As the sample becomes less smooth than the baseline $L_k$, accuracy decreases and accuracy variance increases, yielding more erratic performance. Closer to the model smoothness, accuracy is higher and more stable.
> >
> > #### Generalization study
> >
> > We also conducted fine-grained generalization study in addition to Table 1, where we tested each model on each dataset. Each row is a model trained on that row’s dataset. Each column entry is the test accuracy of the model. We show that even though each dataset concerns covid fake news, classifiers trained on each dataset perform significantly worse on the related datasets, with some notable exceptions.
> >
> > `Table T3`
> >
> > | Row: Training Data | `cov_fn` | `k_short` | `coaid` | `cq`   | `k_long` | `rumor` | `c19_text` | `miscov` | `c19_title` |
> > |--------------------|--------|---------|-------|------|--------|-------|----------|--------|-----------|
> > | `cov_fn`             | **0.96**   | 0.43    | 0.51  | 0.44 | 0.47   | **0.75**  | 0.40      | 0.52   | **0.64**      |
> > | `k_short`            | 0.52   | **0.97**    | 0.56  | 0.57 | **0.86**   | 0.52  | 0.50      | 0.48   | 0.52      |
> > | `coaid`              | 0.20    | 0.53    | **0.97**  | 0.48 | 0.43   | 0.40   | **0.76**     | **0.83**   | **0.84**      |
> > | `cq`                 | 0.57   | 0.57    | 0.55  | **0.54** | 0.59   | 0.42  | 0.55     | 0.41   | 0.46      |
> > | `k_long`             | 0.53   | **0.71**    | 0.58  | 0.4  | **0.98**   | 0.43  | 0.54     | 0.48   | 0.54      |
> > | `rumor`              | **0.67**   | 0.53    | 0.55  | 0.52 | **0.62**   | **0.83**  | 0.48     | 0.34   | **0.60**       |
> > | `c19_text`           | 0.57   | **0.72**    | **0.79**  | 0.44 | **0.67**   | 0.58  | **0.98**     | 0.43   | 0.43      |
> > | `miscov`             | 0.46   | 0.45    | 0.47  | 0.53 | 0.44   | 0.54  | 0.46     | **0.55**   | 0.54      |
> > | `c19_title`          | 0.57   | 0.55    | **0.77**  | 0.55 | **0.62**   | **0.81**  | **0.62**     | 0.43   | **0.95**      |
> >
> >
> > The diagonal bold represents model's accuracy on its test set and is the highest value in each row. However, models also perform well on other datasets (non-diagonal bold) when this test data that matches their training data distribution, or if they are from similar sources. This is the crux of MiDAS: automatically identifying which model best fits a specific sample by examining the neighborhood of the embedding space and model output and comparing smoothness to the training data smoothness. This also address why half-sources give poor accuracy and why increased diversity increases accuracy: we are limiting the training data space. When we increase to all data sources, the increased diversity means we have access to models that are smooth around the test sample. For example, the `coaid` model performs well on `c19_text`, `miscov`, and `c19_title`.
> >
> >
> > Response continued

---

> > > ### Author Response · Authors · 2022-11-19
> > > **Cont. Response to Reviewer 35Bs**
> > >
> > > #### The intended scope of the method; Does it imply any restrictions on the data or the models involved
> > >
> > > We believe MiDAS can be used as a weak-labeling tool. Given multiple source models, each with noisy predictions on some target domain, MiDAS can select the best-fit subset of source models to predict labels, with abstentions from remaining classifiers on a per-sample basis. Labels generated through MiDAS are less noisy, as we showed in Table R2 above: with MiDAS, using only labels inside the smoothness threshold, we have minimal variance in accuracy, whereas relying on the model for all samples without considering local smoothness yields lower *and* noisier accuracy.
> > >
> > > With respect to restrictions on data or models, we presented one such restriction in the feature extractor backbone, where we have mentioned we used foundation models due to their smoother embeddings. We can relax this constraint by requiring instead that MiDAS-compatible classifiers be trained with a smoothness enforcing loss on the latent representation, e.g. GANs with 1-Lipschitzness.
> > >
> > >
> > > ### It will be great if the author could compared their work with the work from the reference above.
> > >
> > > While the work in Chen et al. (2022) could be considered prior work, we believe we arrived at these approaches independently. As such, we have referenced their work to show some overlap. However, there are some key differences between our approaches: we apply a randomized Lipschitzness test; instead of KMeans directly on the embeddings with C classes, where C = number of prediction classes, we use multiple proxies for each class. Since the embeddings are not globally smooth but locally smooth, we cluster into C*p clusters, where p is a constant, e.g. 10; this value can be determined with ELBOW or any off-the-shelf optimal cluster mechanism. So, instead of 2 clusters for fake/true detection, we would use 20 clusters during the clustering step, and compute a local L threshold for each cluster. Finally, we have focused MiDAS on fake news detection as highly relevant to current research. As such, we had not provided a comparison to Chen's approach. We performed a preliminary comparison on the text-datasets in Chen, and show competitive results; ours are slightly better in some cases because we used randomized Lipschitz and multiple proxies for each classification label as clusters. We used BERT as our foundation model.
> > >
> > > | Dataset         |  Chen (2022) |    MiDAS |
> > > |-----------------|--------------|----------|
> > > | Spam (Youtube)  |         0.95 |     0.96 |
> > > | Weather         |         0.98 |     0.98 |
> > > | Spouse          |         0.52 |     0.55 |

---

### Decision · Program_Chairs · 2023-01-20

**Decision:**

Reject

**Justification For Why Not Higher Score:**

The paper chose a specific technical approach to a specific task without good justifications. All reviewers agree that this paper is not ready to be published at ICLR.

**Justification For Why Not Lower Score:**

Reject is the lowest category.

**Metareview: Summary, Strengths And Weaknesses:**

In this paper, the authors propose MIDAS, a statistical machine-learning approach for fake news detection that ranks the relevancy of existing models to new samples. It has a domain-invariant encode and an adaptive module. The authors have empirically verified the model on 9 COVID fake news datasets. The main complaints from reviewers are: (1) The link between the particular method and the COVID fake news detection task is weak. It is unclear why the authors have chosen this particular method (local Lipschitz smoothness of the invariant embedding space) for this particular problem (fake news detection) at a particular time (COVID). (2) The motivation for using Lipschitz smoothness for domain generalization is not strong. There were some other minor concerns about clarity and other issues in the experiments. Overall, there is a clear consensus that this paper is not ready to be accepted by ICLR.

**Summary Of Ac-Reviewer Meeting:**

No. It is not a borderline paper.